# Design, Synthesis, and Evaluation of Novel Indole Hybrid Chalcones and Their Antiproliferative and Antioxidant Activity

**DOI:** 10.3390/molecules28186583

**Published:** 2023-09-12

**Authors:** Zuzana Kudličková, Radka Michalková, Aneta Salayová, Marián Ksiažek, Mária Vilková, Slávka Bekešová, Ján Mojžiš

**Affiliations:** 1NMR Laboratory, Institute of Chemistry, Faculty of Science, Pavol Jozef Šafárik University, 040 01 Košice, Slovakia; maria.vilkova@upjs.sk; 2Department of Pharmacology, Faculty of Medicine, Pavol Jozef Šafárik University, 040 01 Košice, Slovakia; radka.michalkova@upjs.sk; 3Department of Chemistry, Biochemistry and Biophysics, University of Veterinary Medicine and Pharmacy in Košice, 041 81 Košice, Slovakia; aneta.salayova@uvlf.sk (A.S.); marian.ksiazek@student.uvlf.sk (M.K.); 4Thermo Fisher Scientific, 821 09 Bratislava, Slovakia; slavka.bekesova@thermofisher.com

**Keywords:** indole chalcone, antiproliferative activity, antioxidant activity, 2-fluoro substitution, hydroxy substitution

## Abstract

The synthesis, anticancer, and antioxidant activities of a series of indole-derived hybrid chalcones are reported here. First, using the well-known Claisen–Schmidt condensation method, a set of 29 chalcones has been designed, synthesized, and consequently characterized. Subsequently, screening for the antiproliferative activity of the synthesized hybrid chalcones was performed on five cancer cell lines (HCT116, HeLa, Jurkat, MDA-MB-231, and MCF7) and two non-cancer cell lines (MCF-10A and Bj-5ta). Chalcone **18c**, bearing 1-methoxyindole and catechol structural features, exhibited selective activity against cancer cell lines with IC_50_ values of 8.0 ± 1.4 µM (Jurkat) and 18.2 ± 2.9 µM (HCT116) and showed no toxicity to non-cancer cells. Furthermore, antioxidant activity was evaluated using three different methods. The in vitro studies of radical scavenging activity utilizing DPPH radicals as well as the FRAP method demonstrated the strong activity of catechol derivatives **18a**–**c**. According to the ABTS radical scavenging assay, the 3-methoxy-4-hydroxy-substituted chalcones **19a**–**c** were slightly more favorable. In general, a series of 3,4-dihydroxychalcone derivatives showed properties as a lead compound for both antioxidant and antiproliferative activity.

## 1. Introduction

Chalcones are a group of naturally occurring compounds with diverse biological effects that serve as promising starting points for drug design. They possess the structure of 1,3-diphenylprop-2-en-1-one (**1**) and are open-chain precursors for the biosynthesis of flavonoids and isoflavonoids, which are predominantly polyphenolic compounds. Chalcone-containing plants have been extensively utilized in traditional medicine. Additionally, they serve as lead compounds in the development of new drugs [1], with some of them already undergoing clinical trials or being used as drugs [2].

By replacing the phenyl group in the structure of chalcones with various heterocycles, numerous hybrid chalcones with significant biological effects have been obtained [3]. Indole, a potent pharmacodynamic nucleus found in various natural products such as signal molecules (neurotransmitter serotonin, auxin indole-3-acetic acid, indole phytoalexins) as well as many drugs (sunitinib, indomethacin, vincristine, panobinostat) [4], proved to be a suitable heterocycle in the development of hybrid chalcones. One important indole chalcone is MOMIPP (**2**), which effectively reduces the growth and viability of temozolomide-resistant glioblastoma and doxorubicin-resistant breast cancer cells at low micromolar concentrations. This compound was found to act through a non-apoptotic mechanism of cell death—metuosis—and may serve as a prototype drug against cancer that is resistant to common forms of cell death (for example, apoptosis) [5].

In our recent study, among the 19 investigated chalcones with 1-methoxyindole and 2-alkoxyindole skeletons, four inhibited the proliferation of colorectal cancer cells HCT-116 with IC_50_ values < 8 μM and displayed low cytotoxicity towards the fibroblast cell line 3T3. Most of them showed activity against human leukemic T cell lymphoma (Jurkat) with an IC_50_ below 15 μM. The study also demonstrated the binding interactions of selected chalcones with CT DNA and BSA [6]. The chalcone L1 (**3**) was subsequently shown to have anti-proliferative and pro-apoptotic effects against the HeLa cervical cancer model and was also suggested to be a modulator of the anti-tumor microenvironment [7].

Our previous study demonstrated the effectiveness of fluorinated derivatives [7]. Consequently, we are committed to developing and exploring new fluorinated derivatives of chalcones with an indole core. Our determination is supported by the fact that, over the last thirty years, fluoropharmaceuticals have accounted for 20% of all approved medicines [8]. Numerous reviews have been published analyzing the database of fluorinated drugs to provide reliable insights into drug discovery [8,9,10]. In recent years, while the decline of small-molecule drugs has been observed at the expense of the rising star of biologics, the number of fluorine-containing drugs remains comparable to biologics. The number of fluorinated pharmaceuticals is expected to increase in the future in parallel with advances in fluorinated functionalization methodologies [8]. The high prevalence of fluorinated drugs can be attributed to several factors [8] that influence the absorption, distribution, metabolism, and excretion (ADME) of candidate drugs [8]. Halogenation is a commonly employed and successful method for derivatizing chalcones in the development of prototype drug structures. A series of 2-fluoro, 4-fluoro, and 2,5-difluoro-substituted chalcone derivatives were synthesized and evaluated against five cancer cell lines (IC_50_ in the range of 0.029–0.729 µM). Compound **4** was identified as the most promising compound for the development of new therapeutic agents, specifically for the treatment of kidney cancer in humans [11]. In another study, 2-fluoro-4′-aminochalcone **5** was among the most active compounds investigated. It induced apoptosis rather than necrosis in both cells and increased p53 expression in ER-positive cells (MCF-7 line) [12].

As mentioned earlier, chalcones are natural substances widely distributed in vegetables, tea, and other plants. Hydroxyl and methoxy groups are the most commonly used functional groups by nature in the formation of chalcones [13]. Significant findings have emerged from the latest examination of drugs containing hydroxyl groups, which constitute a substantial 37% of all medications [14]. Natural products account for a quarter of all medicines sold, with the majority being of synthetic origin. Up to 69% of natural products contain hydroxyl groups, whereas for synthetics, it is only 23%. It was found that the probability of drugs containing hydroxyl functions is much higher for drugs derived from natural products compared to synthetic drugs origin [14]. Many natural chalcones bearing hydroxyl groups have a wide spectrum of biological activities [15]. Licochalcon A (**6**), isolated from the root of *Glycyrrhiza glabra* (liquorice) or *Glycyrrhiza inflata*, exhibits prominent anticancer effects and has also been found to inhibit the efflux of antineoplastic drugs from cancer cells [15]. Its potential anticancer properties have been demonstrated in various types of cancer cells, including gastric, ovarian, breast, glioma, bladder, and liver cancer cells [16]. Studies show that licochalcone A (**6**) induces apoptosis in U87 glioma cells, nasopharyngeal cancer cells, epithelial ovarian carcinoma cells, and bladder cancer cells.

The 2,4,3′,4′-tetrahydroxychalcone, butein (**8**), inhibits the growth of various breast cancer cells, but the molecular mechanisms underlying butein-induced apoptosis remain unclear. Butein has been reported to induce ROS generation in triple-negative MDA-MB-231 breast cancer cells [17]. However, another study described the reduction of ROS levels leading to apoptosis without affecting butein-resistant HER2^+^ (HCC-1419, SKBR-3, and HCC-2218) cells [18]. Naturally occurring polyhydroxychalcones such as broussochalcone A, butein (**8**), and xanthohumol (**7**), the main prenylchalcone found in hops and beer, appear to be even more potent antioxidants than α-tocopherol [19,20,21].

In a constant effort to identify potential candidates with antitumor properties, our research group has successfully synthesized indole-hybrid chalcones in our laboratory [6,7,22]. Continuing this research, new indole chalcones were designed, synthesized, and investigated for antiproliferative as well as antioxidant activity. The indole core is preserved in all compounds as an active pharmacophoric fragment, and the benzene ring is substituted with a fluoro or hydroxyl group (Figure 1).

## 2. Results and Discussion

### 2.1. Chemistry

The investigated chalcones, numbered **11**–**14** and **17**–**19,** were synthesized using either base (50% aq. KOH or piperidine) or acid-catalyzed (SOCl_2_ in anhydrous ethanol) Claisen–Schmidt reactions according to Figure 1. The 1,3-diarylpropenones with a 3-indol-3-yl moiety **11a**–**c**–**14a**–**c** were prepared by condensation of 1-H (**9a**), 1-methyl (**9b [23]**), or 1-methoxy (**9c**) substituted indole-3-carboxaldehyde [24,25] with equimolar amounts of different acetophenones (2-fluoro-(**10a**)/4-trifluoromethyl-(**10b**)/2-hydroxy-(**10c**) or 4-hydroxy-(**10d**)), as shown in Figure 1. During the KOH-catalyzed condensation of 1-methoxyindole-3-carboxaldehyde (**9c**) with 2-fluoroacetophenone (**10a**), when the reaction time was extended to 24 h, in addition to the condensation reaction, a nucleophilic substitution occurred on the indole nucleus at position 2. This resulted in the formation of 2-alkoxychalcones **11d**–**k**. The alkoxy group binding (depending on the solvent used) occurred simultaneously with the cleavage of the methoxy group from the indole nitrogen. This nucleophilic substitution phenomenon was previously described by Somei [26], and we have recently employed it in the synthesis of similar 2-alkoxyindol-3-yl-4-fluorophenylprop-2-en-1-ones [6].

The reaction of 1-H (**15a**), 1-methyl (**15b [27]**), or 1-methoxy-3-acetylindole (**15c [26]**) with 4-hydroxybenzaldehyde (**16a**), 3,4-dihydroxybenzaldehyde (**16b**), or vanillin (**16c**) resulted in the formation of 1-(indol-3-yl)-1,3-diarylpropenones **17a**–**c**, **18a**–**c**, and **19a**–**c**. Yields ranged from 55 to 81% (Figure 2).

The desired 1,3-diarylpropenones were isolated, purified, and characterized using NMR, IR, and HR-MS. The ^1^H NMR spectra exhibited two diagnostic doublet signals corresponding to H-2 and H-3 protons of 1,3-diarylpropenones, resonating between chemical shift values of 7.02–7.82 ppm and 7.84–8.21 ppm (for 3-(indol-3-yl)prop-2-en-1-ones), and within a narrow range of 7.47–7.64 ppm for both signals in the case of 1-(indol-3-yl)prop-2-en-1-ones. The coupling constant, *J*, for these doublets, ranged from 15.2 to 15.8 Hz, as shown in Table 1. Such large coupling constant values indicate the *trans* (*E*) geometry at the double bond of the propenone linkage. For compounds **17b**,**c**; **18c**; and **19b**,**c**, a strong splitting was observed, where Δν/*J* is less than one. To determine the correct chemical shift positions for peaks H-2 and H-3, we calculated the weighted average of the positions of the two lines, weighted by the intensities of the two lines [28]. Even in the case of compound **18b**, we observed that the H-2 and H-3 protons are isochronous, and the interaction constants could not be determined. The ^13^C NMR spectra of the prepared chalcones displayed signals at the farthest downfield, ranging from 193.1 to 183.1 ppm, assignable to C-1 of the ropanone linkage. The carbonyl atom with 2-hydroxyphenyl substitution (chalcones **13a**–**c**) was shifted downfield to 193.1–193.0 ppm, and C-1 with indol-3-yl substitution (chalcones **17–19**) was shifted upfield to 183.1–183.8 ppm. Signals for C-1, C-2, and C-3 were detected as doublets with *J*_CF_ in the range of 2.3–0.9 Hz for 2-fluorophenyl chalcones **11a**–**k**. A complete assignment of all ^1^H and ^13^C resonances was performed using 2D NMR experiments. ^1^H and ^13^C NMR spectra of synthesized compounds are presented in Appendix A.

### 2.2. Antiproliferative Activity

Screening for antiproliferative activity of the synthesized hybrid chalcones was conducted using five different cancer cell lines: HCT116 (human colorectal carcinoma), HeLa (human cervical adenocarcinoma), Jurkat (human acute T-lymphoblastic leukemia), MDA-MB-231 (human mammary gland adenocarcinoma), and MCF7 (human breast adenocarcinoma). The compounds were evaluated for selective cytotoxicity against cancer cells in comparison to normal cells using two cell lines: MCF-10A (human mammary epithelial cells) and Bj-5ta (immortalized foreskin fibroblasts). The determined IC_50_ values of the investigated chalcones in comparison with the reference drug cisplatin are listed in Table 2. Values lower than 10 μM, as well as values lower than those for cisplatin, are highlighted in bold. Each value represents the mean ± SD of three independent experiments.

The results indicate that the 2-fluoroderivatives **11a**–**k** exhibit moderate activity against cancer cells, with the highest efficacy observed against Jurkat leukemic cells. Chalcones having N-H, N-methoxy, and 2-ethoxy substituents on the indole moiety, such as **11a**, **11c,** and **11e**, exhibited an activity of less than 8.3 μM on this cell line, which is similar to the effectiveness of cisplatin. These compounds exhibited greater activity against cancer cells compared to non-cancer cells, as indicated by their selectivity index [IC_50_(MCF-10A)/IC_50_(Jurkat cells)]. Generally, compounds **11a**, **11c**, and **11e** had a higher selectivity index than 11. In contrast, cisplatin had a selectivity index of only 4. On the other hand, the introduction of the 2-isopropoxy group (chalcone **11g**) decreased selectivity towards non-tumor cells, consistent with our previous study [6]. Chalcone with a 2-propoxy group, **11f**, significantly suppressed the proliferation of breast cancer cells (34 and 37.3 μM for MDA-MB-231 and MCF-7) with minimal effect on non-cancer mammary epithelial cells, MCF-10A. The mode of cell death was recently evaluated by our group [29] using flow cytometry, Western blot, and fluorescence microscopy. Chalcone **11f** was found to induce cell cycle arrest in the G2/M phase and apoptosis. This was associated with the release of cytochrome *c*, increased activity of caspase 3 and caspase 7, PARP cleavage, decreased mitochondrial membrane potential, and the activation of the DNA damage response system. Chalcone **11f** was shown to initiate autophagy as a defense mechanism in treated cells trying to escape the harmful effects induced by the chalcone [29].

Among the 4-trifluoromethyl chalcones **12a**–**c**, selective activity against HCT116 and MCF-7 cells was observed, with an IC_50_ of 12.3 and 20.5 μM for the N-methyl derivative **12b** and 59.1 and 43.4 μM for the unsubstituted derivative **12a**. Minimal activity was observed against non-cancer cells MCF-10A and Bj-5ta (>100 μM). However, such desired selectivity was not observed for the compound with a 1-methoxyindole core **12c**; even on non-cancer MCF-10A cells, higher toxicity was observed than on cancer cells. In a recent study, Lagu et al. [19] synthesized and evaluated the antibacterial and antifungal activity of fluorinated chalcones bearing trifluoromethyl and trifluoromethoxy substituents. Compounds bearing the indole ring **12a** were found to exhibit the greatest antimicrobial activity compared to standard drugs without showing cytotoxicity on a normal human liver cell line (L02). In recent years, the trend has been to develop multi-targeted drugs to fight cancer and microbial infections commonly seen in immunocompromised patients undergoing chemotherapy. Compounds containing a trifluoromethylphenyl group are undoubtedly an excellent choice for designing multi-targeted drugs.

Another evaluated series comprises hydroxylated chalcones **13**–**14a**–**c** and **17**–**19a**–**c**. The results of the antiproliferative activity of the investigated compounds align with the findings that synthetic compounds based on natural substances have a higher likelihood of succeeding as drugs. It is common for natural products to exhibit multiple hydroxyl functions, while most synthetic drugs contain at most one OH group [14]. Natural chalcones often contain a hydroxyl group in position 4, and 3,4-dihydroxy substitution is relatively common in chalcones. Among the evaluated compounds, 4-hydroxyphenyl and 3,4-dihydroxyphenyl chalcones **14c** and **18c**, both bearing a methoxy group on the indole nitrogen, exhibited IC_50_ values of 7.3 and 8.0 µM on the Jurkat cell line, proving to be the most effective. In addition, chalcone **18c** also demonstrated activity against HCT116 with an IC_50_ of 18.5 µM and showed no toxicity against non-cancer cells at the investigated concentrations. The activity of 3,4-dihydroxyphenyl chalcone **18c** is comparable to that of *cis*-Pt on these two cell lines. Plants from the family *Cruciferae* (*Brassicaceae*) produce natural antimicrobial substances called indole phytoalexins, often with the methoxy group attached to the nitrogen in the indole nucleus. In addition to antimicrobial activity, antiproliferative and chemopreventive activities have also been demonstrated for these compounds [30]. The suitability of the methoxy group as a substituent on indole nitrogen for the development of indole hybrid chalcones with an antiproliferative effect was demonstrated in our previous studies [6,7]. Most hydroxylated chalcones evaluated exhibited moderate activity and lacked selectivity. Some, including **14b**, **17a**, **17b**, **18a**, and **19a**–**c**, were more toxic to non-cancer cells than to cancer cells. 

In conclusion, based on the results obtained, establishing a significant correlation between structural features and anticancer activity can be challenging since the activity in vitro varies across different cell lines.

From the findings, it can be inferred that the antiproliferative activity is positively influenced by the methoxy group present on the indole nitrogen and the 2-fluoro- or 3,4-dihydroxy substitution on the phenyl nucleus. Moreover, this effect is accompanied by an increase in selectivity activity between cell lines. Based on its selectivity, compound **18c** appears to be a promising drug candidate among indole hybrid chalcones for targeting human colorectal carcinoma (HCT116) and Jurkat cells.

Furthermore, it is well known that fluorine acts as a weak hydrogen bond acceptor and can serve as a bioisoster of the hydroxyl group (OH) [8]. Our research has revealed that the 2-fluoro compounds **11** tested in this study, as well as the 4-fluoro compounds examined in a previous study [6], exhibited significantly improved biological effects compared to their 2- and 4-hydroxy counterparts **13** and **14**. Further studies on the synthesis of 3,4-difluorophenyl derivatives of indole chalcones would be valuable in determining whether they possess increased activity, similar to the effects observed in our investigation of 3,4-dihydroxylated derivatives **18**.

### 2.3. Antioxidant Activity

In addition to antiproliferative activity, a series of indole-chalcone hybrids were tested for antioxidant activity. Chalcones are precursors for flavonoids and isoflavonoids with well-known antioxidant properties. Additionally, chalcones and chalcone derivatives, especially hydroxy-functionalized chalcones, have shown a high capacity for the scavenging of free radicals [31,32]. Within this study, as seen in Table 3, the set of synthesized hybrid chalcones was assessed for antioxidant activity using three different in vitro methods, namely DPPH (2,2-diphenyl-1-picrylhydrazyl) radical scavenging, ABTS (2,2′-azinobis-(3- ethylbenzothiazoline-6-sulfonic acid) radical scavenging, and ferric reducing antioxidant power assay (FRAP). In addition to chalcones, the activities of the natural polyphenols caffeic acid and *p*-coumaric acid were also determined at the same concentration of 1 mM, and results were expressed as the µmol equivalent of gallic acid per mmol sample. Generally, in vitro antioxidant assays are relatively simple to perform, but the reaction mechanism is complex and depends on several factors [33,34]. In the case of FRAP, the mechanism is due to single electron transfer (SET), but in the ABTS and DPPH methods, it is assumed to be mixed SET and hydrogen atom transfer (HAT) mechanisms [33,35].

The DPPH free radical scavenging method is a simple and accepted tool for screening antioxidant activity. According to the DPPH assay, the synthesized chalcones exhibited activity ranging from weak to strong. Among the synthesized compounds, the series of synthesized compounds **11a**–**c**, **13a**–**c**, **14a**–**c**, and **17a**–**c** showed only mild scavenging activity against the DPPH radical. However, synthesized compounds **18a**–**c** were found to be the most potent structures, with activity ranging from 520.1 ± 6.9 to 589.1 ± 8.9 µmol GAE/mmol. This series exhibited activity comparable to that of caffeic acid (690.7 ± 36.6 GAE/mmol) and about half of that of gallic acid.

The ABTS test proved to be more sensitive for the compounds **11a**–**c**, **13a**–**c**, **14a**–**c**, and **17a**–**c**. Additionally, there was not such a significant difference in the activities between individual series of chalcones, and almost all of them showed considerable reducing ability. However, the chalcone series **19a**–**c** with the 4-hydroxy-3-methoxyphenyl function group demonstrated slightly stronger ABTS radical scavenging than series **18a**–**c**, which have a 3,4-dihydroxyl moiety. It is worth noting that the structural feature of the 4-hydroxy-3-methoxyphenyl unit is also present in natural curcumin and synthetic derivatives with potent antioxidant activity [36].

Additionally, the reducing power of newly synthesized hybrid chalcones was measured using the FRAP assay to determine the total antioxidant power. The first set of compounds, including **11a**–**c**, **13a**–**c**, **14a**–**c**, and **17a**–**c**, showed only poor to moderate reducing activity in this method, similar to the DPPH test. However, the second set of synthesized compounds was more effective. The compounds **18a**–**c** demonstrated the best reducing ability, comparable to caffeic acid, followed by chalcones **19a**–**c**.

Moreover, the obtained correlations quantitatively confirm the parallelism between DPPH scavenging activity and reducing power (R = 0.941).

Generally, the structure–activity relationships confirm that free radical scavenging activity and ferric-reducing capacity are mainly related to the number and position of hydroxyl substituents. The presence of two hydroxyl groups arranged in the catechol moiety of derivatives **18a**–**c** revealed the primary influence on antioxidant activity. This finding is also demonstrated by the improved activity of caffeic acid-bearing 3,4-dihydroxy groups compared to *p*-coumaric acid with only one hydroxyl group. Moreover, it is consistent with the literature, where cinnamic acid substituted in the aromatic ring with two hydroxyl groups (i.e., caffeic acid) had a higher antioxidant capacity compared to monohydroxycinnamic acid (*p*-coumaric acid) measured in DPPH, ABTS, CUPRAC, and FRAP assays [37]. Additionally, naturally occurring 3,4-dihydroxy chalcone broussochalcone A inhibited iron-induced lipid peroxidation and DPPH radical formation and exerted potent inhibitory effects on NO production [38]. A possible mechanism begins with the abstraction of the H atom of the 4-hydroxyl group by the DPPH radical, followed by the abstraction of an additional H atom at the 3-OH position to form a quinone structure [19]. According to the published QSAR analysis of a series of 25 chalcones [39], the spatial, structural, and lipophilic properties of the compounds were shown to determine their antioxidant properties. Our study indicates that the activity among 1-H, 1-methyl-, and 1-methoxy substituted indole derivatives was higher in favor of unsubstituted derivatives. It was also observed that isomerism played a lesser role compared to the type and number of substituents, as there was no significant difference within isomeric chalcones **14a**–**c** and **17a**–**c**. Moreover, the position of a hydroxyl group (2-OH or 4-OH) was less important than the number of oxygen atoms in the molecule. This was evident from the fact that both the 2-alkoxy on the indole (**11f**) and the 3-methoxy on the benzene ring contribute (**19a**–**c**) to the antioxidant activity. The effect of the fluorine atom can be analyzed by comparing compounds **11a**–**c** and **13a**–**c**, where a minor decline in activity was reported by compounds **11a**–**c** containing electron-withdrawing fluorine instead of the H-bond donor hydroxyl group. This finding was also observed in other series of synthetic chalcones [32].

## 3. Materials and Methods

### 3.1. Chemistry

#### 3.1.1. General Method and Materials

All chemicals and solvents used in the synthesis were commercially obtained and were used without further purification unless otherwise specified. Reactions were stirred magnetically and monitored by thin-layer chromatography (TLC). TLC was performed on TLC aluminum sheets coated with silica gel 60 F_254_ and visualized with UV fluorescence. Column chromatographic purification was carried out using Silica 60A particle size 40–63 microns (Davisil^®^) with the indicated eluent. Melting points were measured using a digital melting point apparatus (electrothermal) using open glass capillaries and are uncorrected.

NMR spectra were recorded on a VNMRS spectrometer (Varian) operating at 600 MHz for ^1^H and 150 MHz for ^13^C at 299.15 K. Chemical shifts (*δ* in ppm) are given for the internal solvent, DMSO-d_6_. Based on the ^1^H NMR spectra analysis, it has been determined that the synthesized compounds are of high purity and suitable for biological testing (impurities less than 5%).

Infrared spectra were recorded using an IRAffinity-1 FTIR Spectrophotometer (Shimadzu) in the range 4000−500 cm^−1^ using the KBr method (1 mg sample and 150 mg KBr) or with an Avatar FT−IR 6700 spectrometer in the range 4000−400 cm^−1^ with 64 repetitions for a single spectrum using the ATR (attenuated total reflectance) technique. The obtained data were analyzed using Omnic 8.2.0.387 (2010) software.

High-resolution fragmentation spectra were obtained with the use of the Orbitrap Fusion™ Lumos™ Tribrid™ Mass Spectrometer (Thermo Fisher Scientific, Waltham, MA, USA), following the procedure mentioned in our previous publication [40]. Data processing: To confirm the structure of the studied molecules (**11a**–**k** and **12a**–**c**, **13a**–**c**, **14a**–**c**, **18c**, and **19c**), the precursor ions underwent exhaustive fragmentation using various fragmentation techniques (CID, HCD) at multiple collision energies. The resulting collections of fragmentation spectra at each fragmentation level (MS^2^ to MS^4^) facilitated the generation of a comprehensive spectral tree of information (see Appendix A). The acquired data were manually processed using Mass Frontier™ 8.0 software (Thermo Scientific™, Bratislava, Slovakia) in the Curator module. This module employs advanced algorithms to detect any discrepancies between the declared structure precursor and the product MS^n^ fragmentation spectra. The fragments for each structure were obtained using Mass Frontier™ 8.0 software in the Fragments and Mechanisms module, which allows the prediction of in silico fragmentation and suggests a comprehensive fragmentation pathway based on a set of general ionization, fragmentation, and rearrangement rules. Information from the HighChem Fragmentation Library™, which contains around 227,000 mechanisms and is based on a collection of all available scientific journals for mass spectrometry, was also utilized for predicting possible fragments. The fragments and mechanisms tool automatically generates fragments based on a user-supplied molecular structure. If the in silico-generated fragments of a given compound agree with the observed fragments, the ion peaks in the MS^n^ spectra can be annotated. This curation process was applied to every spectrum at every collision energy, for each fragmentation type, at every MS^n^ level, and for each precursor ion. The obtained data were of high quality. Because of this comprehensive data, which contains high-resolution MS/MS and multi-stage MS^n^ spectra acquired at various collision energies using different fragmentation techniques. Moreover, the unequivocal structure was confirmed with mass errors of less than 3 ppm. As a result, these compounds were added to a high-quality mzCloud™ spectral library, which is commercially available (https://www.mzcloud.org, accessed on 10 July 2023) and used for the identification of small molecules using tandem mass spectrometry. The content of the mzCloud™ spectral library is primarily used to identify an unknown compound through a sub-structural search, where the experimental fragmentation spectrum is searched against the mzCloud™ mass spectral database. For prepared compounds, mzCloud™ ID are **11a** (11251), **11b** (11252), **11c** (11232), **11d** (11233), **11e** (11234), **11f** (11235), **11g** (11248), **11 h** (11249), **11i** (11250), **11j** (11253), **11k** (11266), **12a** (11211), **12b** (11212), **12c** (11210), **13a** (11214), **13b** (11215), **13c** (11213), **14a** (11197), **14b** (11198), **14c** (11196), **18c** (11230) and **19c** (11231).

#### 3.1.2. General Procedure (A) Acid-Catalyzed Claisen-Schmidt Condensation

To a solution of acetophenone or the corresponding acetylindole (1 mmol) and the appropriate arylaldehyde (1 mmol) in anhydrous ethanol (10 mL), molecular sieves were added, and then SOCl_2_ (2 mmol) was slowly added. The reaction’s progress was monitored by TLC. After the reaction was complete, the molecular sieves were filtered out, and the reaction mixture was poured into water, followed by extraction with EtOAc. The organic layer was washed with brine, dried over Na_2_SO_4_, evaporated, and the product isolated through chromatography and crystallization.

#### 3.1.3. General Procedure (B1) Base-Catalyzed Claisen–Schmidt Condensation (50% aq. KOH)

To a stirred solution of acetophenone or the corresponding acetylindole (0.5 mmol) in solvent (5 mL), 50% KOH in H_2_O (0.5 mL) was added, followed by the appropriate arylaldehyde (0.5 mmol). The reaction mixture was stirred at room temperature. After the reaction was complete, the mixture was cooled to 0–10 °C and acidified with 1M HCl to pH4. Then, the precipitated product was filtered, washed with H_2_O and cold alcohol, dried, and crystallized. If the product did not precipitate, the mixture was extracted with EtOAc, and the extract was washed with brine. After drying over anhydrous Na_2_SO_4_, the filtrate was evaporated to dryness, and the product was purified by column chromatography on SiO_2_ and then crystallized to yield the chalcones.

#### 3.1.4. General Procedure (B2) Base-Catalyzed Claisen–Schmidt Condensation (Piperidine)

Arylaldehyde (1 mmol) and acetophenone (1 mmol) were dissolved in anhydrous ethanol (5 mL), and piperidine (2 mmol) was added. The reaction mixture was heated at 60–70 °C, and the course of the reaction was monitored by TLC. After the reaction was complete, the mixture was cooled to room temperature. Then, the precipitated product was filtered, washed with ethanol, dried, and either crystallized or chromatographed on SiO_2_.

#### 3.1.5. Synthesis and Characterization of Compounds **11**–**14**, **17**–**19**

(**2*E*)-1-(2″-Fluorophenyl)-3-(1**′***H*-indol-3**′**-yl)prop-2-en-1-one (11a)**

Procedure A: 1.5 h, 23%; Procedure B2: 27 h, 15%; R*_f_* 0.39 (hexane/acetone 2:1); yellow crystals; mp 103–105.8 °C (CH_2_Cl_2_/hexane); ^1^H NMR (600 MHz, DMSO-*d_6_*): δ 11.97 (s, 1H, NH), 8.08 (s, 1H, H-2′), 7.92 (dd, 1H, *J* 7.1, 1.2 Hz, H-4′), 7.91 (dd, 1H, *J* 15.7, 1.2 Hz, H-3), 7.75 (td, 1H, *J* 7.6, 1.7 Hz, H-6″), 7.63 (dddd, 1H, *J* 8.3, 7.2, 5.2, 1.8 Hz, H-4″), 7.51–7.48 (m, 1H, H-7′), 7.39–7.34 (m, 2H, H-3″, H-5″), 7.29 (dd, 1H, *J* 15.7, 2.3 Hz, H-2), 7.25 (td, 1H, *J* 7.4, 1.3 Hz, H-6′), 7.22 (td, 1H, *J* 7.4, 1.3 Hz, H-5′); ^13^C NMR (150 MHz, DMSO-d_6_): δ 188.2 (d, *J*_CF_ 2.1 Hz, C-1), 160.0 (d, *J*_CF_ 249.8 Hz, C-2″), 140.0 (C-3), 137.6 (C-7′a), 134.1 (C-2′), 133.4 (d, *J*_CF_ 8.7 Hz, C-4″), 130.3 (d, *J*_CF_ 3.0 Hz, C-6″), 127.8 (d, *J*_CF_ 14.0 Hz, C-1″), 124.9 (C-3′a), 124.8 (d, *J*_CF_ 3.2 Hz, C-5″), 122.9 (C-6′), 121.4 (C-5′), 120.1 (C-4′), 119.5 (d, *J*_CF_ 4.9 Hz, C-2), 116.5 (d, *J*_CF_ 22.6 Hz, C-3″), 112.6 (C-7′), 112.4 (C-3′); IR (KBr): ν_max_ 3266, 3036, 1609, 1597, 1570, 1508, 1486, 1413, 1313, 1242, 1197, 1162, 1069, 831, 771, 745 cm^−1^; HRMS: *m*/*z* [M+H]^+^: 266.09757 for C_17_H_13_FNO (calcd. 266.09778).

(**2*E*)-1-(2″-Fluorophenyl)-3-(1**′**-metyl-1**′***H*-indol-3**′**-yl)prop-2-en-1-one (11b)**

Procedure B1: EtOH, 27 h, 57%; R*_f_* 0.49 (hexane/acetone 2:1); yellow crystals; mp 117–120 °C (CH_2_Cl_2_/hexane); ^1^H and ^13^C NMR data identical with literature [22]; IR (KBr): ν_max_ 3104, 1646, 1609, 1585, 1560, 1528, 1448, 1375, 1285, 1078, 1025, 975, 776, 740 cm^−1^; HRMS: *m*/*z* [M+H]^+^: 280.11374 for C_18_H_15_FNO (calcd. 280.11322).

(**2*E*)-1-(2″-Fluorophenyl)-3-(1**′**-methoxy-1**′***H*-indol-3**′**-yl)prop-2-en-1-one (11c)** [6]

(**2*E*)-1-(2″-Fluorophenyl)-3-(2**′**-methoxy-1**′***H*-indol-3**′**-yl)prop-2-en-1-one (11d)**

Procedure B1: MeOH, 24 h, 35%; R*_f_* 0.48 (EtOAc/hexane 2:1); yellow crystals; mp 172–173 °C (acetone/hexane); ^1^H NMR (DMSO-*d_6_*, 600 MHz): δ 12.17 (s, 1H, NH), 7.84 (dd, 1H, *J* 15.4, 1.5 Hz, H-3), 7.70 (td, 1H, *J* 7.6, 1.8 Hz, H-6″), 7.66 (d, 1H, *J* 7.6 Hz, H-4′), 7.59 (dddd, 1H, *J* 8.3, 7.2, 5.2, 1.8 Hz, H-4″), 7.36–7.32 (m, 3H, H-7′, H-5″, H-3″), 7.17 (td, 1H, *J* 7.5, 1.2 Hz, H-6′), 7.12 (td, 1H, *J* 7.7, 1.2 Hz, H-5′), 7.02 (dd, 2H, *J* 15.4, 2.2 Hz, H-2), 4.14 (s, 3H, OCH_3_); ^13^C (DMSO-d_6_, 150 MHz): δ 187.4 (d, *J*_CF_ 2.1 Hz, C-1), 159.8 (d, *J*_CF_ 249.0 Hz, C-2″), 157.7 (C-2′), 136.8 (C-3), 132.9 (d, *J*_CF_ 8.5 Hz, C-4″), 132.1 (C-7′a), 130.2 (d, *J*_CF_ 3.2 Hz, C-6″), 128.2 (d, J_CF_ 14.5 Hz, C-1″), 125.1 (C-3′a), 124.7 (d, *J*_CF_ 3.3 Hz, C-5″), 121.8 (C-5′), 121.2 (C-6′), 118.5 (C-4′), 116.4 (d, *J*_CF_ 22.9 Hz, C-3″), 115.7 (d, *J*_CF_ 5.0 Hz, C-2), 111.6 (C-7′), 93.4 (C-3′), 58.8 (OCH_3_); IR (KBr): ν_max_ 3146, 2938, 1612, 1588, 1560, 1547, 1491, 1474, 1455, 1445, 1434, 1358, 1319, 1285, 1265, 1230, 1211, 1166, 1075, 828, 760 cm^−1^; HRMS: *m*/*z* [M+H]^+^: 296.10846 for C_18_H_15_ FNO_2_ (calcd. 296.10813).


**(2*E*)-3-(2′-ethoxy-1′*H*-indol-3′-yl)-1-(2″-fluorophenyl)prop-2-en-1-one (11e)**


Procedure B1: EtOH, 24 h at 60 °C, 31%; R*_f_* 0.38 (hexane/acetone 2:1); yellow crystals; mp 170–171 °C (acetone/hexane); ^1^H (DMSO-d_6_, 600 MHz): δ 12.11 (s, 1H, NH), 7.88 (dd, 1H, *J* 15.4, 1.5 Hz, H-3), 7.73 (td, 1H, J 7.6, 1.8 Hz, H-6″), 7.66 (d, 1H, *J* 7.6 Hz, H-4′), 7.59 (dddd, 1H, *J* 8.3, 7.2, 5.2, 1.8 Hz, H-4″), 7.36–7.32 (m, 3H, H-7′, H-5″, H-3″), 7.16 (td, 1H, *J* 7.7, 1.2 Hz, H-6′), 7.11 (td, 1H, *J* 7.6, 1.2 Hz, H-5′), 7.07 (dd, 1H *J* 15.4, 2.3 Hz, H-2), 4.45 (q, 2H, *J* 7.0 Hz, CH_2_CH_3_), 1.44 (t, 3H, *J* 7.0, CH_2_CH_3_); ^13^C NMR (150 MHz, DMSO-d_6_): δ 187.1 (d, *J*_CF_ 2.2 Hz, C-1), 160.0 (d, *J*_CF_ 249.2 Hz, C-2″), 156.9 (C-2′), 136.8 (C-3), 133.0 (d, *J*_CF_ 8.7 Hz, C-4″), 132.1 (C-7′a), 130.3 (d, *J*_CF_ 3.2 Hz, C-6″), 128.2 (d, *J*_CF_ 14.2 Hz, C-1″), 125.0 (C-3′a), 124.7 (d, *J*_CF_ 3.3 Hz, C-5″), 121.8 (C-5′), 121.2 (C-6′), 118.4 (C-4′), 116.4 (d, *J*_CF_ 23.0 Hz, C-3″), 115.6 (d, *J*_CF_ 5.5 Hz, C-2), 111.5 (C-7′), 93.9 (C-3′), 67.6 (OCH_2_ CH_3_), 14.7 (OCH_2_CH_3_); IR (KBr): ν_max_ 3179, 2984, 2938, 1611, 1543, 1486, 1386, 1345, 1284, 1262, 1227, 1125, 1099, 1075, 1055, 1008, 828, 774, 747 cm^−1^; HR-MS: *m*/*z* [M+H]^+^: 310.12396 for C_19_H_17_FNO_2_ (calcd. 310.12378).

(**2*E*)-1-(2**″**-Fluorophenyl)-3-(2**′**-propoxy-1**′***H*-indol-3**′**-yl)prop-2-en-1-one (11f)**

Procedure B1: *n*-PrOH, 24 h, 33%; R*_f_* 0.50 (hexane/acetone 2:1); orange crystals; mp 142–144 °C (acetone/hexane); ^1^H NMR (DMSO-d_6_, 600 MHz): δ 12.11 (s, 1H, NH), 7.88 (dd, 1H, *J* 15.4, 1.5 Hz, H-3), 7.71 (td, 1H, *J* 7.6, 1.8 Hz, H-6″), 7.66 (d, 1H, *J* 7.8 Hz, H-4′), 7.59 (tdd, 1H, *J* 8.1, 5.2, 1.8 Hz, H-4″), 7.35–7.32 (m, 3H, H-7′, H-5″, H-3″), 7.16 (td, 1H, *J* 7.5, 1.1 Hz, H-6′), 7.11 (td, 1H, *J* 7.7, 1.1 Hz, H-5′), 7.06 (dd, 1H, *J* 15.4, 2.2 Hz, H-2), 4.36 (t, 2H, *J* 6.5 Hz, OCH_2_CH_2_CH_3_), 1.82 (qt, 2H, J 7.4, 6.5 Hz, OCH_2_CH_2_CH_3_), 1.01 (t, 3H, *J* 7.4 Hz, OCH_2_CH_2_CH_3_); ^13^C NMR (DMSO-d_6_, 150 MHz): δ 187.3 (d, *J*_CF_ 2.2 Hz, C-1), 159.9 (d, *J*_CF_ 249.1 Hz, C-2″), 156.9 (C-2′), 136.9 (C-3), 133.0 (d, *J*_CF_ 8.7 Hz, C-4″), 132.0 (C-7′a), 130.3 (d, *J*_CF_ 3.2 Hz, C-6″), 128.2 (d, *J*_CF_ 14.4 Hz, C-1″), 125,0 (C-3′a), 124.7 (d, *J*_CF_ 3.2 Hz, C-5″), 121.8 (C-5′), 121.2 (C-6′), 118.4 (C-4′), 116.4 (d, *J*_CF_ 23.0 Hz, C-3″), 115.7 (d, *J*_CF_ 5.5 Hz, C-2), 111.5 (C-7′), 93.8 (C-3′), 73.0 (OCH_2_CH_2_CH_3_), 22.1 (OCH_2_CH_2_CH_3_), 10.0 (OCH_2_CH_2_CH_3_). IR (KBr): ν_max_ 1624, 1607, 1579, 1561, 1517, 1484, 1460, 1339, 1280, 1229, 1211, 1203, 1065 cm^−1^; HRMS: *m*/*z* [M+H]^+^: 324.13989 for C_20_H_19_FNO_2_ (calcd. 324.13943).

(**2*E*)-1-(4**″**-Fluorophenyl)-3-(2**′**-isopropoxy-1**′***H*-indol-3**′**-yl)prop-2-en-1-one (11g)**

Procedure B1: *iso*-PrOH, 24 h, 19%; R*_f_* 0.31 (hexane/acetone 2:1) and (hexane/EtOAc 2:1); orange-brown powder; mp 150–152 °C (acetone/hexane); ^1^H (DMSO-d_6_, 600 MHz): δ 12.05 (*s*, 1H, NH), 7.87 (dd, 1H, *J* 15.6, 1.5 Hz, H-3), 7.73 (td, 1H, *J* 7.6, 1.8 Hz, H-6″), 7.67 (d, 1H, *J* 7.8 Hz, H-4′), 7.60 (dddd, 1H, *J* 8.3, 7.3, 5.2, 1.8 Hz, H-4″), 7.36–7.32 (m, 3H, H-7′, H-5″, H-3″), 7.16 (td, 1H, *J* 7.6, 1.1 Hz, H-6′), 7,12 (td, 1H, *J* 7.6, 1.1 Hz, H-5′), 7.09 (dd, 1H, *J* 15.6, 2.3 Hz, H-2), 4,91 (septet, 1H, *J* 6.1 Hz, OCH(CH_3_)_2_), 1.40 (d, 6H, *J* 6.0, OCH(CH_3_)_2_); ^13^C NMR (DMSO-d_6_, 150 MHz): δ 187.2 (d, *J*_CF_ 2.1 Hz, C-1), 160.0 (d, *J*_CF_ 249.1 Hz, C-2″), 155.9 (C-2′), 136.9 (C-3), 133.1 (d, *J*_CF_ 8.6 Hz, C-4″), 132.2 (C-7′a), 130.3 (d, *J*_CF_ 3.0 Hz, C-6″), 128.2 (d, *J*_CF_ 14.6 Hz, C-1″), 124.8 (C-3′a), 124.7 (d, *J*_CF_ 3.0 Hz, C-5″), 121.7 (C-5′), 121.4 (C-6′), 118.5 (C-4′), 116.4 (d, *J*_CF_ 23.0 Hz, C-3″), 115.8 (d, *J*_CF_ 6.0 Hz, C-2), 111.5 (C-7′), 95.2 (C-3′), 75.7 (OCH(CH_3_)_2_), 22.1 (OCH(CH_3_)_2_); IR (KBr): ν_max_ 3179, 2984, 2938, 1611, 1543, 1486, 1386, 1345, 1284, 1262, 1227, 1125, 1099, 1075, 1055, 1008, 828, 774, 747 cm^−1^; HRMS: *m*/*z* [M+H]^+^: 324.13986 for C_20_H_19_FNO_2_ (calcd. 324.13943).

(**2*E*)-3-(2**′**-butoxy-1**′***H*-indol-3**′**-yl)-1-(2**″**-fluorophenyl)prop-2-en-1-one (11h)**

Procedure B1: *n*-BuOH, 24 h, 33%; R*_f_* 0.47 (hexane/acetone 2:1) and (hexane/EtOAc 2:1); dark-orange crystals; mp 141–143 °C (acetone/hexane); ^1^H (DMSO-d_6_, 600 MHz): δ 12.11 (s,1H, NH), 7,86 (dd, 1H, *J* 15.4, 1.5 Hz, H-3), 7.71 (td, 1H, *J* 7.5, 1.7 Hz, H-6″), 7.66 (d, 1H, *J* 7.6 Hz, H-4′), 7.59 (dddd, 1H, *J* 8.3, 7.2, 5.2, 1.8 Hz, H-4″), 7.35–7.31 (m, 3H, H-7′, H-5″, H-3″), 7.16 (td, 1H, *J* 7.6, 1.2 Hz, H-6′), 7.11 (td, 1H, *J* 7.6, 1.2 Hz, H-5′), 7.05 (dd, 1H, *J* 15.4, 2.2 Hz, H-2), 4.39 (t, 2H, *J* 6.4 Hz, CH_2_CH_2_CH_2_CH_3_), 1.81–1.76 (m, 2H, OCH_2_CH_2_CH_2_CH_3_), 1.50–1.44 (m, 2H, OCH_2_CH_2_CH_2_CH_3_), 0.95 (t, 3H, *J* 7.4 Hz, OCH_2_CH_2_CH_2_CH_3_); ^13^C NMR (DMSO-d_6_, 150 MHz): δ 187.4 (d, *J*_CF_ 2.0 Hz, C-1), 159.9 (d, *J*_CF_ 249.1 Hz, C-2″), 156.9 (C-2′), 136.9 (C-3), 132.9 (d, *J*_CF_ 8.7 Hz, C-4″), 132.0 (C-7′a), 130.2 (d, *J*_CF_ 3.1 Hz, C-6″), 128.2 (d, *J*_CF_ 14.4 Hz, C-1″), 125.0 (C-3′a), 124.7 (d, *J*_CF_ 3.2 Hz, C-5″), 121.8 (C-5′), 121.2 (C-6′), 118.4 (C-4′), 116.3 (d, *J*_CF_ 23.0 Hz, C-3″), 115.8 (d, *J*_CF_ 4.9 Hz, C-2), 111.5 (C-7′), 93.9 (C-3′), 71.4 (OCH_2_CH_2_CH_2_CH_3_), 30.6 (OCH_2_CH_2_CH_2_CH_3_), 18.5 (OCH_2_CH_2_CH_2_CH_3_), 13.5 (OCH_2_CH_2_CH_2_CH_3_); IR (KBr): ν_max_ 3052, 2958, 2872, 1622, 1607, 1560, 1524, 1482, 1344, 1280, 1228, 1071, 1015, 982, 775, 739 cm^−1^; HRMS: *m*/*z* [M+H]^+^: 338.15570 for C_21_H_21_FNO_2_ (calcd. 338.15508).

(**2*E*)-3-(2**′**-isobutoxy-1**′***H*-indol-3**′**-yl)-1-(4**″**-fluorophenyl)prop-2-en-1-one (11i)**

Procedure B1: *iso*-BuOH, 24 h, 27%; R*_f_* 0.48 (hexane/acetone 2:1); dark-orange crystals; mp 157–159 °C (acetone/hexane); ^1^H NMR (DMSO-d_6_, 600 MHz): δ 12.11 (s, 1H, NH), 7.88 (dd, 1H, *J* 15.4, 1.4, H-3), 7.70 (td, 1H, *J* 7.4, 1.5 Hz, H-6″), 7.66 (d, 1H, *J* 7.7 Hz, H-4′), 7.59 (dddd, 1H, *J* 8.1, 7.2, 5.3, 1.8 Hz, H-4″), 7.35–7.31 (m, 3H, H-7′, H-5″, H-3″), 7.16 (td, 1H, *J* 7.6, 1.0 Hz, H-6′), 7.11 (td, 1H, *J* 7.6, 1.1 Hz, H-5′), 7.06 (dd, 1H, *J* 15.4, 2.1 Hz, H-2), 4.18 (d, 2H, *J* 6.5 Hz, OCH_2_CH(CH_3_)_2_), 2.16–2.07 (m, 1H, OCH_2_CH(CH_3_)_2_), 1.01 (d, *J* 6.4 Hz, OCH_2_CH(CH_3_)_2_); ^13^C NMR (DMSO-d_6_, 150 MHz): δ 187.4 (d, *J*_CF_ 2.0 Hz, C-1), 159.9 (d, *J*_CF_ 248.9 C-2″), 156.9 (C-2′), 137.0 (C-3), 132.9 (d, *J*_CF_ 8.6 Hz, C-4″), 132.0 (C-7′a), 130.2 (d, *J*_CF_ 3.2 Hz, C-6″), 128.2 (d, *J*_CF_ 14.5 Hz, C-1″), 125.1 (C-3′a), 124.7 (d, *J*_CF_ 3.3 Hz, C-5″), 121.8 (C-5′), 121.2 (C-6′), 118.3 (C-4′), 116.3 (d, *J*_CF_ 22.0 Hz, C-3″), 115.8 (d, *J*_CF_ 5.1 Hz, C-2), 111.5 (C-7′), 93.7 (C-3′), 77.3 (OCH_2_CH(CH_3_)_2_), 27.9 (OCH_2_CH(CH_3_)_2_), 18.6 (OCH_2_CH(CH_3_)_2_); IR (KBr): ν_max_ 3120, 2960, 2872, 1622, 1609, 1558, 1524, 1483, 1459, 1346, 1280, 1230, 1203, 1072, 1017, 829, 775 cm^−1^; HRMS: *m*/*z* [M+H]^+^: 338.15530 for C_21_H_21_FNO_2_ (calcd. 338.15508).


**(2*E*)-1-(2-fluorophenyl)-3-{2-[(1-methoxypropan-2-yl)oxy]-1H-indol-3-yl}prop-2-en-1-one (11j)**


Procedure B1: 1-methoxypropan-2-ol, 24 h, 32%; R*_f_* 0.24 (hexane/EtOAc 2:1); orange oil; ^1^H NMR (DMSO-d_6_, 600 MHz): δ 12.06 (s, 1H, NH), 7.87 (dd, 1H, *J* 15.5, 1.4, H-3), 7.73 (td, 1H, *J* 7.6, 1.8 Hz, H-6″), 7.67 (d, 1H, *J* 7.7 Hz, H-4′), 7.60 (dddd, 1H, *J* 8.2, 7.6, 5.2, 1.8 Hz, H-4″), 7.37–7.31 (m, 3H, H-7′, H-5″, H-3″), 7.16 (td, 1H, *J* 7.5, 1.2 Hz, H-6′), 7.12 (td, 1H, *J* 7.6, 1.2 Hz, H-5′), 7.09 (dd, 1H, *J* 15.5, 2.2 Hz, H-2), 4.91 (tk, 1H, *J* 4.8, 6.3 Hz, OCH(CH_3_)(CH_2_OCH_3_)), 3.58 (d, 2H, *J* 4.8, OCH(CH_3_)(CH_2_OCH_3_)), 3,30 (s, 3H, OCH(CH_3_)(CH_2_OCH_3_)), 1.35 (d, 3H, *J* 6.3 Hz, OCH(CH_3_)(CH_2_OCH_3_)); ^13^C NMR (DMSO-d_6_, 150 MHz): δ 187.7 (d, *J*_CF_ 2.2 Hz, C-1), 160.4 (d, *J*_CF_ 249.3 C-2″), 156.5 (C-2′), 137.4 (C-3), 133.5 (d, *J*_CF_ 8.7 Hz, C-4″), 132.6 (C-7′a), 130.7 (d, *J*_CF_ 3.1 Hz, C-6″), 128.6 (d, *J*_CF_ 14.2 Hz, C-1″), 125.2 (C-3′a), 125.1 (d, *J*_CF_ 3.3 Hz, C-5″), 122.1 (C-5′), 121.8 (C-6′), 119.0 (C-4′), 116.8 (d, *J*_CF_ 23.0 Hz, C-3″), 116.4 (d, *J*_CF_ 5.9 Hz, C-2), 111.9 (C-7′), 95.6 (C-3′), 78.4 (OCH(CH_3_)(CH_2_OCH_3_)), 75.1 (OCH(CH_3_)(CH_2_OCH_3_)), 59.0 (OCH(CH_3_)(CH_2_OCH_3_)), 17.2 (OCH(CH_3_)(CH_2_OCH_3_)); IR: ν_max_ 3191, 2927,1622, 1715, 1607, 1519, 1448, 1335,1275, 1202, 1100, 1060, 1008, 971, 845, 827, 738 cm^−1^; HRMS: *m*/*z* [M+H]^+^: 354.15040 for C_21_H_20_FNO_3_ (calcd. 354.15000).


**(2*E*)-1-(2-fluorophenyl)-3-[2-(2-hydroxyethoxy)-1H-indol-3-yl]prop-2-en-1-one (11k)**


Procedure B1: ethan-1,2-diol, 24 h, 23%; R*_f_* 0.72 (hexane/acetone 1:1) and (hexane/acetone 2:1); orange crystals; mp 140–143 °C (acetone/hexane); ^1^H NMR (DMSO-d_6_, 600 MHz): δ 12.07 (s, 1H, NH), 7.94 (dd, 1H, *J* 15.5, 1.3, H-3), 7.74 (td, 1H, *J* 7.6, 1.8 Hz, H-6″), 7.66 (d, 1H, *J* 7.8 Hz, H-4′), 7.59 (dddd, 1H, *J* 8.5, 7.1, 5.1, 1.8 Hz, H-4″), 7.36–7.32 (m, 3H, H-7′, H-5″, H-3″), 7.16 (td, 1H, *J* 7.5, 1.2 Hz, H-5′), 7.11 (td, 1H, *J* 7.6, 1.2 Hz, H-6′), 7.08 (dd, 1H, *J* 15.5, 2.3 Hz, H-2), 5.14 (s, 1H, OH), 4.43–4.40 (m, 2H, CH_2_), 3.82–3.79 (m, 2H, CH_2_); ^13^C NMR (DMSO-d_6_, 150 MHz): δ 187.0 (d, *J*_CF_ 2.3 Hz, C-1), 160.0 (d, *J*_CF_ 249.4 C-2″), 157.2 (C-2′), 136.8 (C-3), 133.0 (d, *J*_CF_ 8.6 Hz, C-4″), 132.1 (C-7′a), 130.3 (d, *J*_CF_ 3.1 Hz, C-6″), 128.3 (d, *J*_CF_ 14.1 Hz, C-1″), 124.9 (C-3′a), 124.7 (d, *J*_CF_ 3.3 Hz, C-5″), 121.8 (C-6′), 121.2 (C-5′), 118.5 (C-4′), 116.4 (d, *J*_CF_ 23.0 Hz, C-3″), 115.5 (d, *J*_CF_ 5.6 Hz, C-2), 111.5 (C-7′), 93.9 (C-3′), 73.4 (CH_2_), 59.4 (CH_2_); IR: ν_max_ 3054, 1603;1552, 1520, 1479, 1367, 1338, 1275, 1228, 1068, 1051, 1017, 884, 842, 738 cm-^1^; HRMS: *m*/*z* [M+H]^+^: 326.11877 for C_19_H_17_FNO_3_ (calcd. 326.11870).


**(2*E*)-3-(1*H*-indol-3-yl)-1-(4-trifluoromethylphenyl) prop-2-en-1-one (12a)**


Procedure B2: r.t., 24 h, 20%; R*_f_* 0.39 (hexane/EtOAc 2:1); yellow crystals; m.p. 195–197 °C (CH_2_Cl_2_/hexane); m.p. 62 °C [41]; ^1^H NMR (DMSO-d_6_, 600 MHz): δ 11.99 (s, 1H, NH), 8.29 (d, 2H, *J* 8.1 Hz, H-2″,6″), 8.16 (s, 1H, H-2′), 8.11 (d, 1H, *J* 15.4 Hz, H-3), 8.11–8.09 (m, 1H, H-4′), 7.92 (d, 2H, *J* 8.1 Hz, H-3″,5″), 7.63 (d, 1H, *J* 15.4 Hz, H-2), 7.52–7.49 (m, 1H, H-7′), 7.26 (ddd, 1H, *J* 8.6, 7.1, 1.5 Hz, H-6′), 7.24 (ddd, 1H, *J* 8.5, 7.1, 1.4 Hz, H-5′); ^13^C NMR (150 MHz, DMSO-*d*_6_): δ 188.2 (C-1), 141.9 (C-1″), 140.3 (C-3), 137.6 (C-7′a), 134.1 (C-2′), 131.7 (q, *J*_CF_ 32.3 Hz, C-4″), 128.9 (C-2″,6″), 125.6 (q, *J*_CF_ 3.6 Hz, C-3″,5″), 125.1 (C-3′a), 124.0 (q, *J*_CF_ 272.8 Hz, CF_3_), 122.9 (C-6′), 121.3 (C-5′), 120.5 (C-4′), 115.1 (C-2) 112.8 (C-3′), 112.5 (C-7′); IR: ν_max_ 3218, 1630, 1583, 1537, 1512, 1433, 1322, 1242,1228,1108, 1064, 1011, 815, 740 cm^−1^; HRMS *m*/*z*: [M+H]^+^: 316.09442 for C_18_H_12_F_3_NO (calc. 316.09438).


**(2*E*)-3-(1-methyl-1*H*-indol-3-yl)-1-(4-trifluoromethylphenyl) prop-2-en-1-one (12b)**


Mechanochemical synthesis [22] R*_f_* 0.53 (hexane/EtOAc 2:1); yellow crystals; m.p. 182–184 °C; m.p. 160–162 °C [42]; ^1^H and ^13^C NMR data identical with literature [22]; IR: ν_max_ 3110, 1648, 1548, 1524, 1460, 1372, 1330,1319,1107, 1027, 810, 734 cm^−1^; HRMS *m*/*z*: [M+H]^+^: 330.11038 for C_19_H_14_F_3_NO (calc. 330.11003).


**(2*E*)-3-(1-methoxy-1*H*-indol-3-yl)-1-(4-trifluoromethylphenyl) prop-2-en-1-one (12c)**


Procedure B1: 20 min., 25%; Procedure B2: r.t., 24 h, 30%; R*_f_* 0.58 (hexane/EtOAc 2:1); yellow crystals; m.p. 113–115 °C; ^1^H NMR (400 MHz, DMSO-*d*_6_): δ 8.55 (s, 1H, H-2′), 8.30 (d, 2H, *J* 8.1 Hz, H-2″,6″), 8.15 (dt, 1H, *J* 8.0, 0.9 Hz, H-4′), 8.02 (d, 1H, *J* 15.4 Hz, H-3), 7.93 (d, 2H, *J* 8.1 Hz, H-3″,5″), 7.69 (d, 1H, *J* 15.4 Hz, H-2), 7.59 (dt, 1H, *J* 8.2, 0.9 Hz, H-7′), 7.38 (ddd, 1H, *J* 8.2, 7.1, 1.0 Hz, H-6′), 7.31 (ddd, 1H, *J* 8.1, 7.1, 1.1 Hz, H-5′), 4.17 (s, 3H, OCH_3_); ^13^C NMR (100 MHz, DMSO-*d*_6_): δ 188.6 (C-1), 142.1 (C-1″), 139.2 (C-3), 132.9 (C-7′a), 130.2 (C-2′), 132.4 (q, *J*_CF_ 31.8 Hz, C-4″), 129.4 (C-2″,6″), 126.1 (q, *J*_CF_ 3.7 Hz, C-3″,5″), 122.4 (C-3′a), 124.4 (q, *J*_CF_ 272.6 Hz, CF_3_), 124.2 (C-6′), 122.6 (C-5′), 121.2 (C-4′), 117.0 (C-2), 109.6 (C-7′), 109.0 (C-3′), 67.2 (OCH_3_); IR: ν_max_ 3102, 2949, 1655, 1584, 1562, 1556, 1512, 1368, 1317, 1274, 1245, 1212, 1105, 953, 804, 730 cm^−1^; HRMS *m*/*z*: [M+H]^+^: 346.10530 for C_19_H_14_F_3_NO_2_ (calc. 346.10494).


**(2*E*)-1-(2-hydroxyphenyl)-3-(1*H*-indol-3-yl)prop-2-en-1-one (13a)**


Procedure B2: 10.5 h, 57%; R*_f_* 0.39 (hexane/EtOAc 2:1); orange-yellow crystals; m.p. 179.5–181 °C (EtOAc/hexane), 181–185 °C [43]; 165 °C [44]; ^1^H (600 MHz, DMSO-*d_6_*): δ 13.16 (*s*, 1H, OH), 12.07 (*s*, 1H, NH), 8.25 (*dd*, 1H, J 8.1, 1.6 Hz, H-6″), 8.21 (*s*, 1H, H-2′), 8.21 (*d*, 1H, J 15.2 Hz, H-3), 8.15–8.12 (*m*, 1H, H-4′), 7.77 (*d*, 1H, J 15.2 Hz, H-2), 7.56 (*ddd*, 1H, J 8.3, 7.1, 1.6 Hz, H-4″), 7.52–7.50 (*m*, 1H, H-7′), 7.27 (*dd*, 1H, J 7.1, 1.7 Hz, H-6′), 7.25 (*dd*, 1H, J 7.1, 1.6 Hz, H-5′), 7.01 (*ddd*, 1H, 8.1, 7.1, 1.2 Hz, H-5″), 6.98 (*dd*, 1H, J 8.3, 1.1 Hz, H-3″); ^13^C NMR (150 MHz, DMSO-*d_6_*): δ 193.1 (C-1), 162.2 (C-2″), 140.4 (C-3), 137.6 (C-7a′), 135.7 (C-4″), 134.4 (C-2′), 130.3 (C-6″), 125.1 (C-3a′), 123.0 (C-6′), 121.5 (C-5′), 120.52 (C-1″), 120.5 (C-4′), 119.0 (C-5″), 117.7 (C-3″), 113.9 (C-2), 113.0 (C-3′), 112.6 (C-7′); IR: (KBr) ν_max_ 3299, 3092, 1631, 1577, 1547, 1484, 1438, 1373, 1344, 1293, 1249, 1202, 1153, 1108, 1032, 966, 830, 762, 740 cm^−1^; HRMS: *m*/*z* [M + H]^+^: 264.101912 for C_17_H_13_NO_2_ (calcd. 264.10220).


**(2*E*)-1-(2-hydroxyphenyl)-3-(1-methyl-1*H*-indol-3-yl)prop-2-en-1-one (13b)**


Procedure B: 7 h; 90%; 15% [45]; R*_f_* 0.37 (hexane/EtOAc 2:1); yellow crystals; m.p. 205–207 °C (MeCN), 208–209 °C [45]; ^1^H (600 MHz, DMSO-*d_6_*): δ 13.16 (*s*, 1H, OH), 8.24 (*dd*, 1H, J 8.1, 1.6 Hz, H-6″), 8,20 (*s*, 1H, H-2′), 8.16 (*d*, 1H, J 15.2 Hz, H-3), 8.15 (*d*, 1H, J 8.0 Hz, H-4′), 7.75 (*d*, 1H, J 15.2 Hz, H-2), 7.59 (*d*, 1H, J 8.0 Hz, H-7′), 7.54 (*ddd*, 1H, J 8.3, 7.1, 1.6 Hz, H-4″), 7.34 (*ddd*, 1H, J 8.0, 7.0, 1.1 Hz H-6′), 7.30 (*ddd*, 1H, J 8.0, 7.1, 1.1 Hz, H-5′), 7.01 (*ddd*, 1H, J 8.1, 7.1, 1.2 Hz, H-5″), 6.98 (*dd*, 1H, 8.3, 1.2 Hz, H-3″), 3.88 (*s*, 3H, CH_3_); ^13^C NMR (150 MHz, DMSO-*d_6_*): δ 193.0 (C-1), 162.2 (C-2″), 139.7 (C-3), 138.1 (C-7a′), 137.7 (C-2′), 135.7 (C-4″), 130.3 (C-6″), 125.6 (C-3a′), 123.0 (C-6′), 121.8 (C-5′), 120.6 (C-4′), 120.5 (C-1″), 119.0 (C-5″), 117.7 (C-3″), 113.9 (C-2), 112.0 (C-3′), 111.0 (C-7′), 33.2 (N-CH_3_); IR: (KBr) ν_max_ 3097, 3058, 2909, 2833, 1632, 1573, 1548, 1528, 1487, 1393, 1346, 1298, 1262, 1205, 1135, 1075, 1029, 842, 773, 755 cm^−1^; HRMS: *m*/*z* [M + H]^+^: 278.11768 for C_18_H_15_NO_2_ (calcd. 278.11765).


**(2*E*)-1-(2-hydroxyphenyl)-3-(1-methoxy-1*H*-indol-3-yl)prop-2-en-1-one (13c)**


Procedure B (50 °C): 3.5 h; 53%; R*_f_* 0.62 (hexane/EtOAc 2:1); light-orange crystals; m.p. 139–140 °C (CH_2_Cl_2_/hexane); ^1^H (600 MHz, DMSO-*d_6_*): δ 13.0 (*s*, 1H, OH), 8.59 (*s*, 1H, H-2′), 8.24 (*dd*, 1H, J 8.1, 1.6 Hz, H-6″), 8.17 (*d*, 1H, J 7.9 Hz, H-4′), 8.11 (*d*, 1H, J 15.3 Hz, H-3), 7.82 (*d*, 1H, J 15.3 Hz, H-2), 7,60 (*d*, 1H, J 8.1 Hz, H-7′), 7.55 (*ddd*, 1H, J 8.3, 7.1, 1.6 Hz, H-4″), 7.38 (*t*, 1H, J 7.6 Hz H-6′), 7.32 (*t*, 1H, J 7.6 Hz, H-5′), 7.02 (*dd*, 1H, J 8.1, 7.1 Hz, H-5″), 6.99 (*d*, 1H, 8.3 Hz, H-3″), 4.18 (*s*, 3H, OCH_3_); ^13^C NMR (150 MHz, DMSO-*d_6_*): δ 193.1 (C-1), 162.1 (C-2″), 138.7 (C-3), 135.8 (C-4″), 132.5 (C-7a′), 130.4 (C-6″), 129.9 (C-2′), 123.8 (C-6′), 122.3 (C-5′), 122.1 (C-3a′), 120.8 (C-4′), 120.5 (C-1″), 119.0 (C-5″), 117.8 (C-3″), 115.7 (C-2), 109.2 (C-7′), 108.7 (C-3′), 66.8 (OCH_3_); IR: (KBr) ν_max_ 3462, 3103, 1632, 1560, 1488, 1377, 1352, 1298, 1261, 1204, 1156, 1034, 953, 840, 762, 724 cm^−1^; HRMS: *m*/*z* [M + H]^+^: 294.11292 for C_18_H_15_NO_3_ (calcd. 294.11247).


**(2*E*)-1-(4-hydroxyphenyl)-3-(1*H*-indol-3-yl)prop-2-en-1-one (14a)**


Procedure A: 1 h, 18%; 65% [46]; R*_f_* 0.22 (hexane/acetone 2:1); light-yellow crystals; m.p. 218 °C d (acetone/hexane); 147–148 °C [46]; ^1^H (600 MHz, DMSO-*d_6_*): δ 11.84 (*s*, 1H, OH), 10.28 (*s*, 1H, NH), 8.08–8.06 (*m*, 1H, H-4′), 8.07 (*s*, 1H, H-2′), 8.04 (*d*, 2H, J 8.7 Hz, H-2″, H-6″), 7.99 (*d*, 1H, J 15.5 Hz, H-3), 7.64 (*d*, 1H, J 15.5 Hz, H-2), 7.49–7.48 (*m*, 1H, H-7′), 7.25–7.21 (*m*, 2H, H-5′, H-6′), 6.90 (*d*, 2H, J 8.7 Hz, H-3″, H-5″); ^13^C NMR (150 MHz, DMSO-*d_6_*): δ 187.0 (C-1), 161.7 (C-4″), 137.7 (C-3), 137.5 (C-7a′), 132.6 (C-2′), 130.6 (C-2″, C-6″), 129.8 (C-1″), 125.2 (C-3a′), 122.6 (C-6′), 121.0 (C-5′), 120.3 (C-4′), 115.4 (C-2), 115.3 (C-3″, C-5″), 112.8 (C-3′), 112.4 (C-7′); IR: (KBr) ν_max_ 3414, 3293, 3092, 1642, 1604, 1557, 1443, 1348, 1274, 1230, 1167, 1039, 818, 737 cm^−1^; HRMS: *m*/*z* [M + H]^+^: 264.10193 for C_17_H_13_NO_2_ (calcd. 264.10191).


**(2*E*)-1-(4-hydroxyphenyl)-3-(1-methyl-1*H*-indol-3-yl)prop-2-en-1-one (14b)**


Procedure A: 1.5 h, 12%; R*_f_* 0.29 (hexane/acetone 2:1); light-yellow crystals; m.p. 244 °C d (acetone/hexane); ^1^H (600 MHz, DMSO-*d_6_*): δ 10.29 (*s*, 1H, OH), 8.08 (*dt*, 1H, J 8.0, 1.1 Hz, H-4′), 8.05 (*s*, 1H, H-2′), 8.03 (*d*, 2H, J 8.7 Hz, H-2″, H-6″, 7.94 (*d*, 1H, J 15.5 Hz, H-3), 7.62 (*d*, 1H, J 15.5 Hz, H-2), 7.55 (*dt*, 1H, J 8.1, 1.1 Hz, H-7′), 7.31 (*ddd*, 1H, J 8.1, 7.0, 1.1 Hz, H-6′), 7.27 (*ddd*, 1H, J 8.0, 7.0, 1.1 Hz, H-5′), 6.90 (*d*, 2H, J 8.7 Hz, H-3″, H-5″), 3.85 (*s*, 3H, CH_3_); ^13^C NMR (150 MHz, DMSO-*d_6_*): δ 187.0 (C-1), 161.6 (C-4″), 138.0 (C-7a′), 137.0 (C-3), 136.0 (C-2′), 130.6 (C-2″, C-6″), 129.8 (C-1″), 125.6 (C-3a′), 122.7 (C-6′), 121.3 (C-5′), 120.5 (C-4′), 115.4 (C-2), 115.3 (C-3″, C-5″), 111.8 (C-3′), 110.8 (C-7′), 33.0 (CH_3_); IR (KBr): ν_max_ 3097, 1626, 1587, 1528, 1508, 1473, 1387, 1276, 1225, 1165, 1077, 1050, 973, 816, 745 cm^−1^; HRMS: *m*/*z* [M + H]^+^: 278.11752 for C_18_H_15_NO_2_ (calcd. 278.11756).


**(2*E*)-1-(4-hydroxyphenyl)-3-(1-methoxy-1*H*-indol-3-yl)prop-2-en-1-one (14c)**


Procedure A: 4.5 h, 28%; R*_f_* 0.33 (hexane/acetone 2:1); orange crystals; m.p. 182–184 °C (acetone/hexane); ^1^H (600 MHz, DMSO-*d_6_*): δ 10.32 (*s*, 1H, OH), 8.46 (*s*, 1H, H-2′), 8.11 (*dt*, 1H, J 8.0, 1.0 Hz, H-4′), 8.05 (*d*, 2H, J 8.7 Hz, H-2″, H-6″), 7.91 (*d*, 1H, J 15.5 Hz, H-3), 7.69 (*d*, 1H, J 15.5 Hz, H-2), 7.57 (*dt*, 1H, J 8.1, 1.0 Hz, H-7′), 7.36 (*ddd*, 1H, J 8.1, 7.1, 1.0 Hz, H-6′), 7.27 (*ddd*, 1H, J 8.0, 7.1, 1.0 Hz, H-5′), 6.91 (*d*, 2H, J 8.7 Hz, H-3″, H-5″), 4.15 (*s*, 3H, OCH_3_); ^13^C NMR (150 MHz, DMSO-*d_6_*): δ 186.9 (C-1), 161.7 (C-4″), 136.1 (C-3), 132.4 (C-7a′), 130.8 (C-2″, C-6″), 129.6 (C-1″), 128.5 (C-2′), 123.5 (C-6′), 122.0 (C-3a′), 121.8 (C-5′), 120.7 (C-4′), 117.0 (C-2), 115.3 (C-3″, C-5″), 109.0 (C-3′), 108.7 (C-7′), 66.6 (OCH_3_); IR: (KBr) ν_max_ 3241, 1643, 1591, 1555, 1507, 1327, 1287, 1211, 1166, 1046, 952, 838, 742 cm^−1^; HRMS: *m*/*z* [M + H]^+^: 294.11282 for C_18_H_15_NO_3_ (calcd. 294.11247).


**(2*E*)-3-(4-hydroxyphenyl)-1-(1*H*-indol-3-yl)prop-2-en-1-one (17a)**


Procedure A: 7 h, 49%; 55% [46]; Rf = 0.27 (hexane/acetone 2:1); light-orange crystals; m.p. 197–200 °C d (acetone/hexane); 213–215 °C [46]; ^1^H (600 MHz, DMSO-*d_6_*): δ 12.03 (d, 1H, *J* 3.1 Hz, NH), 9.92 (*s*, 1H, OH), 8.66 (*d*, 1H, *J* 3.1 Hz, H-2′), 8.33 (*ddd*, 1H, *J* 7.9, 1.5, 0.8 Hz, H-4′), 7.70–7.67 (m, 2H, H-2″, H-6″), 7.63 (*d*, 1H, J 15.4 Hz, H-3), 7.55 (*d*, 1H, *J* 15.4 Hz, H-2), 7.48 (*dt*, 1H, *J* 7.9, 1.0 Hz, H-7′), 7.23 (*ddd*, 1H, *J* 7.9, 7.0, 1.5 Hz, H-6′), 7.20 (*ddd*, *J* 8.0, 6.9, 1.0 Hz, 1H, H-5′), 6.85–6.81 (m, 2H, H-3″, H-5″); ^13^C NMR (150 MHz, DMSO-*d_6_*): δ 183.8 (C-1), 159.3 (C-4″), 139.8 (C-3), 136.8 (C-7a′), 134.2 (C-2′), 130.3 (C-2″, C-6″), 126.3 (C-1″), 125.9 (C-3a′), 123.0 (C-6′), 121.8 (C-5′), 121.7 (C-4′), 121.3 (C-2), 117.8 (C-3′), 115.7 (C-3″, C-5″), 112.1 (C-7′); IR: ν_max_ 3493, 3120, 1633, 1604, 1584, 1512, 1442, 1238, 1156, 970. 822, 746 cm^−1^.


**(2*E*)-3-(4-hydroxyphenyl)-1-(1-methyl-1*H*-indol-3-yl)prop-2-en-1-one (17b)**


Procedure A: 7 h, 55%; R*_f_* 0.33 (hexane/acetone 2:1); light-red crystals; m.p. 236–239 °C (acetone/hexane); ^1^H (600 MHz, DMSO-*d_6_*): δ 9.94 (*s*, 1H OH), 8.67 (s, 1H, H-2′), 8.34 (*dt*, 1H, *J* 7.9, 1.0 Hz, H-4′), 7.69–7.65 (*m*, 2H, H-2″, H-6″), 7.56 (*dt*, 1H, *J* 8.1, 0.9 Hz, H-7′), 7.56 (*d*, 1H, *J* 15.8 Hz, H-3), 7.56 (*d*, 1H, *J* 15.8 Hz, H-2), 7.30 (*ddd*, 1H, *J* 8.1, 7.1, 1.3 Hz, H-6′), 7.25 (*ddd*, 1H, *J* 7.9, 7.1, 1.0 Hz, H-5′), 6.86–6.83 (m, 2H, H-3″, H-5″), 3.90 (s, 3H, CH_3_); ^13^C NMR (150 MHz, DMSO-*d_6_*): δ 183.3 (C-1), 159.3 (C-4″), 139.8 (C-3), 137.8 (C-2′), 137.4 (C-7a′)130.2 (C-2″, C-6″), 126.3 (C-3a′), 126.2 (C-1″), 123.0 (C-6′), 122.0 (C-5′), 121.9 (C-4′), 121.2 (C-2), 116.6 (C-3′), 115.7 (C-3″, C-5″), 110.6 (C-7′), 33.3 (CH_3_); IR: ν_max_ 3227, 3115, 1634, 1604, 1556, 1523, 1510, 1463, 1428, 1371, 1267, 1217, 1087, 988, 969, 829, 749, cm^−1^.


**(2*E*)-3-(4-hydroxyphenyl)-1-(1-methoxy-1*H*-indol-3-yl)prop-2-en-1-one (17c)**


Procedure A: 5 h, 76%; R*_f_* 0.38 (hexane/acetone 2:1); light-yellow crystals; m.p. 213–215 °C (acetone/hexane); ^1^H (600 MHz, DMSO-*d_6_*): δ 9.97 (*s*, 1H, OH), 9.06 (*s*, 1H, H-2′), 8.38 (*dt*, 1H, *J* 8.0, 1.0 Hz, H-4′), 7.71–7.67 (*m*, 2H, H-2″, H-6″), 7.59 (*d*, 1H, *J* 15.5 Hz, H-3), 7.58 (*dt*, 1H, *J* 8.1, 0.9 Hz, H-7′), 7.57 (*d*, 1H, *J* 15.5 Hz, H-2), 7.36 (*ddd*, 1H, *J* 8.1, 7.1, 1.1 Hz, H-6′), 7.29 (*ddd*, 1H, *J* 8.0, 7.1, 1.0 Hz, H-5′), 6.86–6.83 (*m*, 2H, H-3″, H-5″), 4.21 (*s*, 3H, OCH_3_); ^13^C NMR (150 MHz, DMSO-*d_6_*): δ 183.4 (C-1), 159.5 (C-4″), 140.4 (C-3), 132.1 (C-7a′), 131.0 (C-2′), 130.4 (C-2″, C-6″), 126.1 (C-1″), 123.9 (C-6′), 122.8 (C-5′), 122.5 (C-3a′), 122.2 (C-4′), 120.8 (C-2), 115.7 (C-3″, C-5″), 113.0 (C-3′), 108.8 (C-7′), 66.9 (OCH_3_); IR: ν_max_ 3103, 1633, 1607, 1586, 1510, 1388, 1368, 1325, 1275, 1236, 1192, 1167, 1065, 975, 949, 815, 739, 715 cm^−1^.


**(2*E*)-3-(3,4-dihydroxyphenyl)-1-(1*H*-indol-3-yl)prop-2-en-1-one (18a)**


Procedure A: 3 h, 80%; R*_f_* 0.11 (hexane/acetone 2:1); yellow-green crystals; m.p. 212–215 °C d (acetone/hexane); ^1^H (600 MHz, DMSO-*d_6_*): δ 12.01 (*bs*, 1H, OH), 9.26 (*bs*, 1H, OH), 8.65 (s, 1H, H-2′), 8.32 (*ddd*, 1H, *J* 7.9, 1.4, 0.9 Hz, H-4′), 7.54 (*d*, 1H, *J* 15.5 Hz, H-3), 7.48 (*dt*, 1H, *J* 7.9, 1.0 Hz, H-7′), 7.47 (*d*, 1H, *J* 15.5 Hz, H-2), 7.24–7.21 (*m*, 1H, H-6′), 7.23 (*d*, 1H, *J* 2.1 Hz, H-2″), 7.19 (*ddd*, 1H, *J* 8.1, 7.0, 1.1, H-5′), 7.14 (*dd*, 1H, 8.1, 2.1 Hz), 6.80 (d, 1H, *J* 8.1 Hz, H-5″); ^13^C NMR (150 MHz, DMSO-*d_6_*): δ 183.8 (C-1), 147.8 (C-4″), 145.5 (C-3″), 140.3 (C-3), 136. 8 (C-7′a), 134.1 (C-2′), 126. 8 (C-1″), 125.9 (C-3′a), 122.9 (C-6′), 121.8 (C-5′), 121.6 (C-4′), 121.2 (C-6″), 121.2 (C-2), 117.8 (C-3′), 115.7 (C-5″), 115.4 (C-2″), 112.1 (C-7′); IR: ν_max_ 3150, 1632, 1607, 1583, 1514, 1440, 1290, 1236, 1150, 1099, 984, 958, 840, 808, 749 cm^−1^.


**(2*E*)-3-(3,4-dihydroxyphenyl)-1-(1-methyl-1*H*-indol-3-yl)prop-2-en-1-one (18b)**


Procedure A: 4 h, 81%; R*_f_* 0.15 (hexane/acetone 2:1); yellow-green crystals; m.p. 210–214 °C d (acetone/hexane); ^1^H (600 MHz, DMSO-*d_6_*): δ 9.30 (*s*, 2H, OH), 8.67 (s, 1H, H-2′), 8.33 (*dt*, 1H, *J* 7.9, 1.0 Hz, H-4′), 7.56 (*dt*, 1H, *J* 8.1, 0.9 Hz, H-7′), 7.47 (*s*, 2H, H-3, H-2), 7.30 (*ddd*, 1H, *J* 8.1, 7.1, 1.3 Hz, H-6′), 7.25 (*ddd*, 1H, *J* 8.0, 7.1, 1.1 Hz, H-5′), 7.22 (*d*, 1H, *J* 2.1 Hz, H-2″), 7.11 (*dd*, 1H*, J* 8.2, 2.1 Hz, H-6″), 6.80 (*d*, 1H, *J* = 8.1 Hz, H-5″), 3.90 (s, 3H, CH_3_); ^13^C NMR (150 MHz, DMSO-*d_6_*): δ 183.3 (C-1), 147.8 (C-4″), 145.6 (C-3″), 140.4 (C-3), 137.8 (C-2′), 137.4 (C-7′a), 126.7 (C-1″), 126.3 (C-3′a), 123.0 (C-6′), 122.0 (C-5′), 121.9 (C-4′), 121.2 (C-6″), 121.1 (C-2), 116.6 (C-3′), 115.7 (C-5″), 115.2 (C-2″), 110.6 (C-7′), 33.3 (CH_3_); IR: ν_max_ 3383, 2951, 2715, 1623, 1598, 1518, 1438, 1368, 1276, 1185, 1084, 966, 769 cm^−1^.


**(2*E*)-3-(3,4-dihydroxyphenyl)-1-(1-methoxy-1*H*-indol-3-yl)prop-2-en-1-one (18c)**


Procedure A: 3 h, 74%; R*_f_* 0.21 (hexane/acetone 2:1); brown-grey crystals; m.p. 206–209 °C d (acetone/hexane); ^1^H (600 MHz, DMSO-*d_6_*): δ 9.75–8.90 (*bs*, 2H, OH), 9.07 (*s*, 1H, H-2′), 8.38 (*dt*, 1H, *J* 7.9, 0.9 Hz, H-4′), 7.58 (*dt*, 1H, *J* 8.1, 0.9 Hz, H-7′), 7.51 (*d*, 1H, *J* 15.5 Hz, H-2), 7.50 (*d*, 1H, *J* 15.5 Hz, H-3), 7.35 (*ddd*, 1H, *J* 8.1, 7.1, 1.0 Hz, H-6′), 7,29 (*ddd*, 1H, *J* 7.9, 7.1, 1.0 Hz, H-5′), 7.25 (*d*, 1H, *J* 2.1 Hz, H-2″), 7.14 (*dd*, 1H, *J* 8.1, 2.1 Hz, H-6″), 6.81 (*d*, 1H, *J* 8.1 Hz, H-5″), 4.21 (*s*, 3H, OCH_3_); ^13^C NMR (150 MHz, DMSO-*d_6_*): δ 183.4 (C-1), 148.0 (C-4″), 145.6 (C-3″), 140.9 (C-3), 132.1 (C-7′a), 131.0 (C-2′), 126.7 (C-1″), 123.9 (C-6′), 122.7 (C-5′), 122.5 (C-3′a), 122.2 (C-4′), 121.4 (C-6″), 120.8 (C-2), 115.6 (C-5″), 115.4 (C-2″), 113.0 (C-3′), 108.8 (C-7′), 66.9 (OCH_3_); IR (KBr): ν_max_ 3526, 3441, 3118 do 2560, 1636, 1508, 1450, 1370, 1326, 1283, 1253, 1205, 1113, 1063, 977, 801, 738 cm^−1^; HRMS: *m*/*z* [M+H]^+^: 310.10764 for C_18_H_15_ NO_4_ (calcd. 310.10738).


**(2*E*)-3-(4-hydroxy-3-methoxyphenyl)-1-(1*H*-indol-3-yl)prop-2-en-1-one (19a)**


Procedure A: 4 h, 79%; 75% [47]; R*_f_* 0.20 (hexane/ acetone 2:1); ligth-orange crystals; m.p. 200–203 °C (acetone/hexane); 215 °C [47]; ^1^H (600 MHz, DMSO-*d_6_*): δ 12.05 (*bs*, 1H, NH), 9.51 (*bs*, 1H, OH), 8.69 (s *s*, 1H, H-2′), 8.33 (*dt*, 1H, *J* 7.8, 1.0 Hz, H-4′), 7.64 (*d*, 1H, *J* 15.4 Hz, H-2), 7.55 (*d*, 1H, *J* 15.4 Hz, H-3), 7.49 (*dt*, 1H, *J* 7.9, 1.0 Hz, H-7′), 7.46 (*d*, 1H, *J* 1.8 Hz, H-2″), 7.25–7.22 (m, 1H, H-6′), 7.23 (*dd*, 1H, *J* 8.1, 1.8 Hz, H-6″), 7.20 (*ddd*, 1H, *J* 8.1, 7.1, 1.2 Hz, H-5′), 6.83 (*d*, 1H, *J* 8.1 Hz, H-5″), 3.88 (*s*, 3H, OCH_3_); ^13^C NMR (150 MHz, DMSO-*d_6_*): δ 183.8 (C-1), 148.8 (C-4″), 147.9 (C-3″), 140.2 (C-3), 136.8 (C-7′a), 134.2 (C-2′), 126.7 (C-1″), 125.9 (C-3′a), 123.1 (C-6″), 123.0 (C-6′), 121.8 (C-5′), 121.7 (C-4′), 121.4 (C-2), 117.8 (C-3′), 115.5 (C-5″), 112.1 (C-7′), 111.4 (C-2″), 55.8 (OCH_3_); IR: ν_max_ 3544, 3115, 1638, 1504, 1427, 1279, 1261, 1149, 1137, 975, 745 cm^−1^.


**(2*E*)-3-(4-hydroxy-3-methoxyphenyl)-1-(1-methyl-1*H*-indol-3-yl)prop-2-en-1-one (19b)**


Procedure A: 4.5 h, 81%; R*_f_* 0.30 (hexane/ acetone 2:1); red crystals; m.p. 170–172 °C (acetone/hexane); ^1^H (600 MHz, DMSO-*d_6_*): δ 9.54 (*bs*, 1H, OH), 8.67 (*s*, 1H, H-2′), 8.34 (*dd*, 1H, *J* 7.9, 1.0 Hz, H-4′), 7.56 (*d*, 1H, *J* 15.6 Hz, H-2), 7.58–7.56 (m, 1H, H-7′), 7.55 (*d*, 1H, *J* 15.6 Hz, H-3), 7.42 (*d*, 1H, *J* 1.8 Hz, H-2″), 7.31 (*ddd*, 1H, *J* 8.1, 7.1, 1.0 Hz, H-6′), 7.25 (*ddd*, 1H, *J* 8.0, 7.1, 1.0 Hz, H-5′), 7.23 (*dd*, 1H, *J* 8.1, 1.8 Hz, H-6″), 6.84 (*d*, 1H, *J* 8.1 Hz, H-5″), 3.91 (*s*, 3H, N-CH_3_), 3.88 (*s*, 3H, C-OCH_3_); ^13^C NMR (150 MHz, DMSO-*d_6_*): δ 183.3 (C-1), 148.9 (C-4″), 147.9 (C-3″), 140.3 (C-3), 137.8 (C-2′), 137.5 (C-7′a), 126.7 (C-1″), 126.4 (C-3′a), 123.0 (C-6′), 122.96 (C-6″), 122.0 (C-5′), 121.9 (C-4′), 121.3 (C-2), 116.6 (C-3′), 115.6 (C-5″), 111.5 (C-2″), 110.6 (C-7′), 55.8 (C-OCH_3_), 33.3 (N-CH_3_); IR ν_max_ 3514, 3118, 1616, 1579, 1512, 1400, 1372, 1267, 1207, 1123, 1081, 964, 740 cm^−1^.


**(2*E*)-3-(4-hydroxy-3-methoxyphenyl)-1-(1-methoxy-1*H*-indol-3-yl)prop-2-en-1-one (19c)**


Procedure A: 3 h, 61%; R*_f_* 0.12 (hexane/EtOAc 2:1); ligth-yellow crystals; m.p. 155–156 °C (acetone/hexane); ^1^H NMR (600 MHz, DMSO-*d_6_*): δ 9.57 (*bs*, 1H, OH), 9.07 (*s*, 1H, H-2′), 8.39 (*dd*, 1H, *J* 8.1, 0.9 Hz, H-4′), 7.59 (*d*, 1H, *J* 15.5 Hz, H-2), 7.60–7.58 (m, 1H, H-7′), 7.58 (*d*, 1H, *J* 15.5 Hz, H-3), 7.46 (*d*, 1H, *J* 1.8 Hz, H-2″), 7.36 (*ddd*, 1H, *J* 8.1, 7.1, 1.0 Hz, H-6′), 7.30 (*ddd*, 1H, *J* 8.0, 7.1, 0.9 Hz, H-5′), 7.25 (*dd*, 1H, *J* 8.1, 1.8 Hz, H-6″), 6.84 (*d*, 1H, *J* 8.1 Hz, H-5″), 4.22 (*s*, 3H, N-OCH_3_), 3.89 (*s*, 3H, C-OCH_3_); ^13^C NMR (150 MHz, DMSO-*d_6_*): δ 183.3 (C-1), 149.1 (C-4″), 147.9 (C-3″), 140.9 (C-3), 132.1 (C-7′a), 131.0 (C-2′), 126.6 (C-1″), 123.9 (C-6′), 123.3 (C-6″), 122.8 (C-5′), 122.5 (C-3′a), 122.2 (C-4′), 120.9 (C-2), 115.6 (C-5″), 113.0 (C-3′), 111.5 (C-2″), 108.9 (C-7′), 66.9 (N-OCH_3_), 55.9 (C-OCH_3_); IR (KBr): ν_max_ 3526, 3423, 3255, 1642, 1586, 1519, 1453, 1377, 1329, 1284, 1201, 1126, 1065, 1031, 976, 842, 807, 738 cm^−1^; HR-MS: *m*/*z*: [M+H]+ 324.12323 for C_19_H_17_NO_4_ (calcd. 324.12303).

### 3.2. Antiproliferative Activity Studies

#### 3.2.1. Cell Cultures

In the experiments assessing the biological activity of the synthesized chalcones, various tumor cell lines were used. Cell lines MDA-MB-231 (human breast adenocarcinoma), HeLa (human cervical adenocarcinoma), HCT116 (human colorectal carcinoma), and Jurkat (human T lymphocyte leukemia) were cultured in RPMI 1640 medium (Biosera, Kansas City, MO, USA). MCF-7 (human breast adenocarcinoma) cells were cultured in Dulbecco’s Modified Eagle’s Medium (DMEM). Media were enriched with 1% HyCloneTM antibiotic/antimycotic solution containing penicillin, streptomycin, and amphotericin B (Merck, Darmstadt, Germany) and 10% fetal bovine serum (FBS; Gibco, Thermo Scientific, Rockford, IL, USA). The non-tumor cell line MCF-10A (human mammary epithelial cells) was cultured in DMEM F12 medium (high-glucose Dulbecco’s Modified Eagle’s Medium F12, Biosera, Kansas City, MO, USA), supplemented with antibiotic/antimycotic solution, insulin (final concentration of 10 µg/mL), 10% fetal bovine serum, EGF (final concentration of 20 ng/mL), and hydrocortisone (final concentration of 0.5 µg/mL) (Merck, Darmstadt, Germany). Human fibroblast cells, Bj-5ta (immortalized foreskin fibroblasts), were cultured in a mixture of DMEM and M199 media in a 4:1 ratio and supplemented with 10% FBS and hygromycin B (final concentration of 0.01 mg/mL). Cells were maintained in a humidified atmosphere containing 5% CO_2_ at 37 °C. Cell viability was higher than 95% for each experiment.

#### 3.2.2. MTT Assay

To evaluate the antiproliferative/cytotoxic activity and inhibition of the metabolism of the tested compounds against cell lines, a colorimetric test of the metabolic activity of MTT (3-(4,5-di-methylthiazol-2-yl)2,5-diphenyltetrazolium bromide) was used (Sigma–Aldrich Chemie, Steinheim, Germany). Tested cells were seeded on 96-well culture plates at a density (5 × 103/well) and cultured in the respective culture medium for 24 h. Tested chalcones were added to the cells at concentrations of 10, 50, and 100 μmol/L and incubated for 72 h. After incubation, 100 µL of 10% MTT solution (5 mg/mL, Sigma–Aldrich Chemie, Steinheim, Germany) was added to each well. After 4 h at 37 °C in a 5% CO_2_ atmosphere, MTT precipitated to insoluble formazan, and 100 µL of 10% SDS (sodium dodecyl sulfate) was added to each well. After dissolving the crystals after 12 h, the absorbance was measured at a wavelength of 540 nm using the automated Cytation™ 3 Cell Imaging Multi-Mode Reader (Biotek, Winooski, VT, USA). IC50 values, determined as half-inhibitory concentrations versus control cells, were calculated using the statistical predictive function Trend.

### 3.3. Antioxidant Activity Studies

#### 3.3.1. DPPH Radical Scavenging Activity

All antioxidant activity measurements were performed using a UV–vis spectrophotometer (Shimadzu UV-1280, Japan). The DPPH-scavenging activities of the compounds, based on the decolorization of the stable purple DPPH free radical measured at 517 nm [48], were estimated with slight modification [49]. A total of 50 µL of sample stock solution (1 mmol·L^−1^) or standard gallic acid (0.05–1.5 mmol·L^−1^) was mixed with 2.0 mL of DPPH solution (0.1 mmol·L^−1^). At the same time, 50 µL of methanol was used instead of the sample as a blank. After 30 min of incubation in the dark, absorbance was measured. The inhibition of radicals was calculated as a percentage of the blank according to the formula (%) = (1 − A_sample_/A_blank_) × 100, and then the antioxidant activity was expressed as micromoles of gallic acid equivalents per millimole of sample (µmol GAE/mmol).

#### 3.3.2. ABTS Radical Scavenging Activity

The ABTS radical scavenging ability of the compounds was determined by the decrease in absorbance of the radical cation (ABTS^•+^) at 734 nm. The ABTS^•+^ was generated by reacting equal volumes of ABTS (7 mmol·L^−1^) in H_2_O with the oxidizing agent K_2_S_2_O_8_ (2.45 mmol·L^−1^) for 12–16 h in the dark at room temperature and subsequently diluted with methanol to obtain an absorbance of 0.76 ± 0.01 at 734 nm. Then, 50 µL of sample stock solution (1 mmol·L^−1^) or standard gallic acid (0.05–1.5 mmol·L^−1^) was added to the ABTS solution (2 mL) and mixed. The absorbance of the sample was read at 734 nm after 30 min of incubation at room temperature. The inhibition of radicals was calculated as a percentage of the blank according to the formula (%) = (1 − A_sample_/A_blank_) × 100, and then the antioxidant activity was expressed as micromoles of gallic acid equivalents per millimole of sample (µmol GAE/mmol).

#### 3.3.3. Ferric Reducing Antioxidant Power (FRAP)

The FRAP method is based on the measurement of intense blue-colored Frap-Fe^2+^ complex formation having an absorption maximum at 595 nm after the reduction of Fe^3+^ to Fe^2+^ ions in the complex by sample [50]. The FRAP reagent contained 10 mmol·L^−1^ of TPTZ (2,4,6-tris(2-pyridyl)-s-triazine) solution in 40 mmol·L^−1^ HCl, 20 mM FeCl_3_, and 0.3 M acetate buffer at pH 3.6 in a ratio of 1:1:10. Briefly, 50 µL of sample stock solution (1 mmol·L^−1^) or standard gallic acid (0.05–1.5 mmol·L^−1^) were mixed with 2 mL of FRAP reagent and incubated for 30 min in the dark. The antioxidant activity was expressed as micromoles of gallic acid equivalents per millimole of sample (µmol GAE/mmol).

#### 3.3.4. Statistical Analysis

The experimental results were performed in triplicate. The data were recorded as mean ± standard deviation and analyzed by one-way ANOVA followed by Tukey’s *t*-test. Statistical analyses were performed using GraphPad Prism 5 software (GraphPad Software Inc., San Diego, CA, USA). Results were considered significantly different when *p* < 0.05.

## 4. Conclusions

In summary, we reported the synthesis of a novel series of hybrid chalcones bearing unsubstituted N-metyl- or N-methoxy-indole pharmacophores and benzene rings with OH, F, and CF_3_ substituents. Antiproliferative activity screening indicates that the 2-fluoroderivatives **11a**–**k** exhibit moderate and selective activity against cancer cell lines. The highest efficacy is observed against Jurkat leukemic cells, with even N-H, N-methoxy, and 2-ethoxy chalcones **11a**, **11c**, and **11e** exhibiting activity of less than 8.3 μM on this cell line. Another 2-fluro-chalcone with 2-propoxy group **11f** significantly suppresses the proliferation of breast cancer cells with minimal effect on non-cancer mammary epithelial cells (MCF-10A). The compound (2E)-3-(3,4-dihydroxyphenyl)-1-(1-methoxy-1*H*-indol-3-yl)prop-2-en-1-one (**18c**) from a series of hydroxylated chalcones demonstrated remarkable effectiveness against Jurkat leukemia cells and a human colon cancer cell line HCT116, with IC_50_ values of 8.0 ± 1.4 and 18.2 ± 2.9 μM, respectively. This structure contains a 1-methoxyindole nucleus, which shows great potential for the development of prototype anticancer drugs. Additionally, 1-methoxychalcone **14c** exhibited impressive efficacy against Jurkat leukemia cells with an IC_50_ value of 7.3 ± 0.1 μM. These findings confirm the suitability of integrating the 1-methoxyindole nucleus into the design of novel and effective cancer treatments. However, further studies focused on a more detailed understanding of the antiproliferative mechanism of selected chalcone derivatives, as well as in vivo studies to confirm the anticancer effects of these compounds, will be necessary. Compounds **18a**–**c** and **19a**–**c** possess the highest antioxidant potential as determined by the in vitro DPPH, ABTS, and FRAP methods.

## Data Availability

Not applicable.

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
