# Peer review of "Design, Synthesis, and Evaluation of Novel Indole Hybrid Chalcones and Their Antiproliferative and Antioxidant Activity"

_molecules, 2023, doi:10.3390/molecules28186583_

Round 1

Reviewer 1 Report

The manuscript "Design, synthesis, and evaluation of novel indole hybrid chalcones and their anticancer and antioxidant activity", shows relevant aspects of new compounds based on indole. However, authors should review the following aspects:

- What was the criterion for selecting the cell lines?

- It could not be said that the compounds have anticancer activity. This statement can only be stated if in vivo studies were performed. In this case you can only say antitumor

- The MTT assay determines the cytotoxicity of the compounds. However, it is not a specific test to indicate that the compounds have antitumor activity. The authors should include other experiments or indicate that through the MTT assay the compounds MAY be excellent antitumor agents. And in the future this activity will be confirmed through other tests.

- In biological assays, positive controls (drugs) must be added to compare the results obtained.

The choice of tumor and non-tumor cell lines should be similar. That is, choose cell lines from the breast, cervix, rectal colon, fibroblasts tumor and non-tumor, in order to compare the results obtained.

The manuscript presents several typographical errors resulting from writing in American and British English. Please review the entire manuscript.

Reviewer 2 Report

This is an interesting paper that is generally easy to follow. However, the written English could be improved in places eg to remove grammatical and typographical errors.

The methods have been clearly stated so that others could replicate the results. The compounds have also been well characterised. However the authors do not comment on how they ascertained the purity of the compounds (eg was elemental analysis or HPLC undertaken) and this needs to be clarified and added in the revised manuscript.

The authors should also discuss how the potencies/selectivities of their compounds compare with compounds that are used in the clinic, so that the reader can see whether this study has made progress in this area.

Moreover, the authors should outline whether they anticipate any challenges progressing the study to in vivo studies, eg with respect to metabolism of the compounds, and should suggest some future avenues for research.

This is an interesting paper that is generally easy to follow. However, the written English could be improved in places eg to remove grammatical and typographical errors.

Reviewer 3 Report

The manuscript describes the synthesis of 29 novel indole-chalcone hybrid compounds by known methods. The newly synthesized compounds were screened for their anticancer and antioxidant properties. The authors have already published  relevant articles in this field and this study is an addition to their previously published works. The manuscript is interesting and can be accepted for publication in Molecules after addressing the following comments:

1) The introduction is too long and it will be better if it can be shortened somehow.

2) Even though the authors have explained about the effect of electron-withdrawing and electron-donating substituents in biological activity of synthesized compounds, a separate paragraph about the Structure Activity Relationship studies will add more value to this work.

3) There are some grammatical errors and typos in the manuscript and hence a thorough proof reading is necessary; For example, "tree different methods", "potent antioxidants then α-tocopherol", "simply and an accepted mechanism", "DPHH" (line 290), "Section 3.3: Antiproliferative studies".

The manuscript may be published after addressing these comments. 

Minor errors need to be correcetd.

Round 2

Reviewer 1 Report

The authors of the manuscript "Design, synthesis, and evaluation of novel indole hybrid chalcones and their anticancer and antioxidant activity" have made the pertinent corrections suggested by the reviewers. The manuscript can be published in its current state on the journal's platform.